# Hydrothermal Activation of Porous Nitrogen-Doped Carbon Materials for Electrochemical Capacitors and Sodium-Ion Batteries

**DOI:** 10.3390/nano10112163

**Published:** 2020-10-29

**Authors:** Yuliya V. Fedoseeva, Egor V. Lobiak, Elena V. Shlyakhova, Konstantin A. Kovalenko, Viktoriia R. Kuznetsova, Anna A. Vorfolomeeva, Mariya A. Grebenkina, Alina D. Nishchakova, Anna A. Makarova, Lyubov G. Bulusheva, Alexander V. Okotrub

**Affiliations:** 1Nikolaev Institute of Inorganic Chemistry SB RAS, 3 Acad. Lavrentiev Ave., Novosibirsk 630090, Russia; LobiakEV@niic.sbras.ru (E.V.L.); ShlyakhovaEV@niic.sbras.ru (E.V.S.); k.a.kovalenko@niic.nsc.ru (K.A.K.); v.kuznetsova5@g.nsu.ru (V.R.K.); vorfolomeeva@niic.nsc.ru (A.A.V.); grmariya@mail.ru (M.A.G.); nishchakova@niic.nsc.ru (A.D.N.); spectrum@niic.nsc.ru (A.V.O.); 2Novosibirsk State University, 2 Pirogova Str., Novosibirsk 630090, Russia; 3Physical Chemistry, Institute of Chemistry and Biochemistry, Free University of Berlin, 14195 Berlin, Germany; anna.makarova@fu-berlin.de

**Keywords:** porous nitrogen-doped carbon, hydrothermal treatment, XPS, NEXAFS, electrochemical double-layer capacitors, sodium-ion batteries

## Abstract

Highly porous nitrogen-doped carbon nanomaterials have distinct advantages in energy storage and conversion technologies. In the present work, hydrothermal treatments in water or ammonia solution were used for modification of mesoporous nitrogen-doped graphitic carbon, synthesized by deposition of acetonitrile vapors on the pyrolysis products of calcium tartrate. Morphology, composition, and textural characteristics of the original and activated materials were studied by transmission electron microscopy, X-ray photoelectron spectroscopy, near-edge X-ray absorption fine structure spectroscopy, infrared spectroscopy, and nitrogen gas adsorption method. Both treatments resulted in a slight increase in specific surface area and volume of micropores and small mesopores due to the etching of carbon surface. Compared to the solely aqueous medium, activation with ammonia led to stronger destruction of the graphitic shells, the formation of larger micropores (1.4 nm vs. 0.6 nm), a higher concentration of carbonyl groups, and the addition of nitrogen-containing groups. The tests of nitrogen-doped carbon materials as electrodes in 1M H_2_SO_4_ electrolyte and sodium-ion batteries showed improvement of electrochemical performance after hydrothermal treatments especially when ammonia was used. The activation method developed in this work is hopeful to open up a new route of designing porous nitrogen-doped carbon materials for electrochemical applications.

## 1. Introduction

Growing global demand for energy requires the development of highly performing, inexpensive, and environmentally friendly electrochemical devices. High-energy-density lithium-ion batteries (LIBs) and high-power-density electrochemical double-layer capacitors (EDLCs) are attracting much attention because they are crucial to modern energy technologies [1,2,3,4,5]. Sodium-ion batteries (SIBs) are considered as a promising low-cost alternative for widely used LIBs due to the abundant sodium sources. EDLCs are considered highly complementary to batteries for energy storage mainly due to their high-power density, extremely long cycle life, and more reliable safety. One of the major challenges in the development of electrochemical devices is electrode materials with outstanding performances. In this regard, extensive research is needed to establish the structure–property relationships in electrode material and to search for new approaches to achieve the desired material properties.

Carbon-based nanomaterials such as activated carbon, carbon nanotubes, and graphene are effectively used as electrode materials in EDLCs and alkali-metal batteries [6,7]. Small specific surface area and elongated structure of carbon nanotubes and dense assembly of restacked graphene layers do not allow electrolyte and metal ions to penetrate an entire volume of materials, limiting their rate capability performance. Thus far, thin highly porous carbon is the most successful electrode material for EDLCs and SIBs due to its large specific surface area, developed mesopore and micropore structure, low cost, good electrical conductivity, and high chemical stability [6,8]. N-doped carbon materials are more attractive because the nitrogen functionalities increase microporosity, electrical conductivity, and induce pseudocapacitance. High electronegativity of graphitic nitrogen leads to polarization of carbon environment, while nitrogen atoms located at the edges of graphene planes and defects can participate in Faradaic reactions [5,9,10,11,12,13]. In case of alkali-ion batteries, nitrogen functionalities shorten the diffusion path of alkali ions in porous carbon and reduce the volume expansion of carbon materials [14,15,16,17,18]. Carbon-based energy storage devices have numerous advantages over other energy storage technologies but could realize further gains if their electrode materials would be properly optimized. Physical activation by CO_2_ and chemical activation by KOH, ZnCl_2_, and acids are widely used to create pores in carbon materials [19,20,21,22,23]. However, such treatments require high temperatures and/or aggressive reagents, which cause dramatic loss of carbon material and nitrogen incorporated in the carbon lattice [20,23] that may not satisfy the required characteristics of electrode materials.

This research work is devoted to the study of hydrothermal treatment on the structure and composition of N-doped carbon material and the evaluation of the influence of these effects on their electrochemical behavior when used as electrodes for EDLCs and SIBs. For this purpose, N-doped carbon material consisting of thin graphitic shells and a high volume of mesopores produced using calcium tartrate and acetonitrile was chosen [24]. We used water and diluted ammonium hydroxide, safe solvents, to modify N-doped carbon without the use of irrelevant pre-oxidation step. Hydrothermal modification in water with or without the addition of ammonia at moderate temperatures makes the activation procedure effortless, inexpensive, and ecologically friendly. The morphology, composition, atomic, and pore structure of materials were comprehensively studied by transmission electron microscopy (TEM), X-ray photoelectron spectroscopy (XPS), near-edge X-ray absorption fine structure (NEXAFS) spectroscopy, Fourier transformed infrared (FTIR) spectroscopy, and nitrogen gas adsorption technique. The effect of the used hydrothermal modification on the energy storage performance of N-doped carbon nanomaterial in EDLCs and SIBs was thoroughly investigated.

## 2. Materials and Methods

### 2.1. Synthesis

Porous N-doped carbon material (denoted as N-C) was synthesized by the CVD method using conditions described in details elsewhere [24]. Decomposition of calcium tartrate (CaC_4_H_4_O_6_) at 750 °C yielded carbon-coated CaO nanoparticles, which were used for the deposition of acetonitrile (CH_3_CN) vapors at the same temperature, leading to the formation of graphitic layers. CaO template nanoparticles were removed using a treatment by diluted HCl for 30 min, the product was washed twice with distilled water to neutral pH, and dried in air at 100 °C overnight.

The hydrothermal activation of N-C material was carried out in water or aqueous ammonia (9 wt.%) solution. The N-C sample (50 mg) was placed in a Teflon reactor (Khimprom, Novocheboksarsk, Russia) and water (5 mL) or the ammonia solution (8 mL) was added there. The reactor was placed in an autoclave and then heated at a given temperature for 30 h. Hydrothermal treatment in water was performed at 200 °C, the saturated vapor pressure was ~15 atm. In order to maintain this vapor pressure during hydrothermal treatment in the ammonia solution, the reactor was heated to 147 °C. The synthesis products were dried in an oven at 80 °C for 12 h. The samples, obtained by hydrothermal activation in water and the ammonia solution, are denoted N-Cw and N-Ca, respectively.

### 2.2. Characterization

The samples’ morphology was investigated by TEM on a JEOL 2010 microscope (JEOL Ltd., Tokyo, Japan) at 200 kV acceleration voltage. Functional groups were identified by FTIR spectroscopy on a Fourier spectrometer Scimitar FTS 2000 (Digilab, Holliston, MA, USA) in the range of 400–4000 cm^−1^. A specimen (~1 mg) was pressed in a KBr tablet using the standard procedure. The recorded data were processed by subtracting the nonlinear background arising from optical components and KBr.

XPS and NEXAFS experiment were carried out at the Russian–German Beamline at BESSY II synchrotron radiation facility, Helmholtz-Zentrum Berlin für Materialien und Energie. The XPS spectra were measured using a PHOIBOS 150 electron-energy analyzer (SPECS GmbH, Berlin, Germany) at an energy of monochromatized synchrotron radiation of 830 eV with a resolution of less than 0.5 eV. The energy scale was calibrated using the binding energy of Au 4f_7/2_ component at 84 eV measured from a clean gold foil. After subtraction of a Shirley background, XPS C 1s and N 1s spectra were fitted using line shapes included Gaussian and Lorentzian functions within Casa XPS software, Version 2.3.15 (Casa Software Ltd., Teignmouth, UK). NEXAFS C K-edge spectra were acquired simultaneously in the total electron yield (TEY) and Auger electron yield (AEY) modes in an energy range of 280–310 eV with 0.1 eV step. The XPS and NEXAFS spectroscopy measurements were carried out at a pressure of 10^−10^ Torr and room temperature.

The porous structure was analyzed by a nitrogen adsorption technique using an Autosorb iQ analyzer (Quantachrome Instruments, Boynton Beach, FL, USA) at 77 K. The compound was first activated in a dynamic vacuum using standard “outgas” option of the equipment at 200 °C during 6 h. N_2_ adsorption−desorption isotherms were measured within the range of relative pressures of 10^−6^ to 0.995. The specific surface area was calculated from the data obtained based on the conventional BET (Brunauer, Emmett, and Teller) and DFT (Density Functional Theory) models. Pore size distributions were determined using the DFT approach, which gives good agreement between measured and calculated isotherms (fitting error less than 0.5%).

### 2.3. Electrochemical Measurements

A three-electrode system was used to measure the performance of EDLC. Carbon material was mixed with an aqueous Teflon F-4D suspension and ethanol, the mixture was homogenized and rolled out into a film. The dried film was applied on a Pt current collector. A Pt foil and an Ag/AgCl electrode filled with a saturated 3.5M KCl aqueous solution were used as the counter and reference electrodes, respectively. A 1M H_2_SO_4_ aqueous solution was prepared as an electrolyte. Cyclic voltammetry (CV) curves were recorded on a SP-300 potentiostat/galvanostat (Bio-Logic Science Instrument, Seyssinet-Pariset, France) at different constant scan rates from 2 and 1000 mVs^−1^ in a voltage range from 0 to + 1 V. The long cycling tests were performed using symmetric two-electrode cells at 100 mV s^−1^. The gravimetric capacitance of a working electrode (C) was calculated from the area of CV curves according to the following formula: $C=\frac{\oint \mathrm{idu}}{2\cdot \Delta U\cdot V_{s}\cdot m}$, where i is the current (A), du is the differentially small increment of potential (V), ∆U is the voltage window (V), V_s_ is the scan rate (V s^−1^), and m is the mass of carbon material (g).

Electrochemical tests of SIBs were conducted in CR2032 coin-type cells assembled in an argon-filled glove box with water and oxygen contents less than 1 ppm. Carbon material was mixed with conductive additive super P and polyvinylidene difluoride binder (in a weight ratio of 8:1:1) in N-methyl pyrrolidone solvent to form a homogeneous slurry. The slurry was coated onto a Cu foil and dried at 100 °C for 12 h in a vacuum oven. All working electrodes had a standard area of ~0.8 cm^2^ and weight of the active carbon material of 0.4–0.6 mg. Sodium metal sheet was used as a counter electrode. Glass fiber piece was used as a separator. The electrolyte was a 1M solution of NaClO_4_ in a mixture of ethylene carbonate and dimethyl carbonate (1:1 by volume). The assembled half-cells were galvanostatically charged and discharged between 0.01 and 2.5 V versus Na/Na^+^ at different current densities (50, 100, 250, 500, and 1000 mA g^−1^) at room temperature using a Land CT2001 battery test system (NEWARE, HongKong, China). The SP-300 instrument was used to record the CV curves in a voltage range of 2.5–0.01 V versus Na/Na^+^ at a potential scan rate of 0.5 mV s^−1^ and electrochemical impedance spectra (EIS) measurements from 100 kHz to 10 mHz at an AC amplitude of 5 mV at a cell potential of 2.5 V. CV and EIS measurements were performed using the sodium-ion half-cells after their performance of galvanostatic charge–discharge cycles.

## 3. Results

### 3.1. Structural Aspect

TEM images of the samples before and after hydrothermal treatments are compared in Figure 1. The original N-C sample has a sponge-like structure of carbon skeleton and pores ranging from 5 to 30 nm, which are morphologic replications of the template CaO nanoparticles [24,25] (Figure 1a). High-resolution TEM observation revealed that carbon shells consist of corrugated and disordered graphene-like layers and have a thickness of less than 5 nm (bottom image in Figure 1a).

Decomposition process of the used organic salt of calcium mainly determines the morphology of carbon shells. Similar thin porous carbon shells with a thickness of about 10 nm were fabricated by pyrolysis of sodium citrate used as template and carbon precursors [8]. N-Cw and N-Ca samples retained sponge-like morphology (Figure 1b,c). However, high-resolution TEM images showed that the graphene-like layers became slightly thinner and more defective especially at the edge because of the etching (bottom images in Figure 1b,c).

Survey XPS spectra of samples detected signals from carbon, oxygen, and nitrogen elements. The surface composition estimated from these spectra is CN_0.06_O_0.05_ for N-C, CN_0.05_O_0.05_ for N-Cw, and CN_0.08_O_0.02_ for N-Ca. XPS C 1s spectra were fitted by five components (Figure 2a) and the fitting results are collected in Table 1. Full width at half maximum (FWHM) is ~1.4 eV for the peak at 284.5 eV, assigned to graphite-like sp^2^ carbon, and ~1.9 eV for other components. A high intensity of the 284.5 eV peak indicates that porous carbon is graphitic mainly, but rather high FWHM value of this component reveals the presence of structural disorders and different orientations of C=C bonds. The component at 285.2 eV (denoted as dis.) may indicate the presence of disordered, amorphous, and diamond-like carbon, as well as carbon atoms bonded to hydrogen [26,27,28]. The surface fraction of this disordered carbon in N-C is 19%, and it increases to 23 and 36% in N-Cw and N-Ca, respectively. The components at 286.5 and 287.7 eV can be assigned to carbon atoms bonded with oxygen through single (C–O) and double (C=O) bonds, accordingly [28]. The component at 288.8 eV arises from carboxyl groups. Different bonding states between carbon and nitrogen contribute to the C 1s spectra at energies between 286 and 287 eV [29]. A slight enhance of the component at 287.7 eV and a strong increase of the peak at 285.2 eV were detected in the C 1s spectra of activated samples (Figure 2a and Table 1). More carbonyl groups and defective states were found on the surface of N-Ca sample.

XPS N 1s spectra revealed at least four different chemical states of nitrogen on the surface of samples (Figure 2b). According to the literature, the peaks located at 398.5, 400.1, 401.2, and 402.3 eV correspond to pyridinic (N_pyr_), hydrogenated (N_hyd_), graphitic (N_gr_), and oxygenated (N_ox_) nitrogen, respectively [11,30,31,32]. The quantitative analysis of nitrogen functionalities is given in Table 1. N_pyr_ and N_hyd_ states (38% of each form) are prevalent in the initial sample, which also contains 16% of N_gr_ and 6% of N_ox_. Hydrothermal treatment in water (N-Cw sample) leads mainly to a decrease in N_hyd_ fraction and an increase in N_gr_ fraction. The use of ammonia in the treatment, on the contrary, increases N_hyd_ content and decreases N_gr_ content in N-Ca sample.

Figure 3 compares NEXAFS C K-edge spectra of the samples measured in TEY and AEY modes. AEY probes the outermost layers, while TEY detects more bulk states. All spectra show two main peaks at 285.4 and 292.5 eV, which are respectively assigned to electron transitions from 1s to π*- and σ*-states in graphitic-like structures [33,34,35]. Broad and smoothed π*- and σ*-resonances in the spectra of N-C sample are a sign of disordering of graphitic layers and this disorder is larger in the surface carbon shells probed by the AEY (Figure 3b) than in depth of the sample (the TEY spectrum in Figure 3a). The spectra of N-Cw and N-Ca show suppression of π*-resonance. This phenomenon indicates a decrease of the size of conjugated π-system due to the etching of the graphitic layers or their functionalization. The strongest changes of carbon state are found for N-Ca (Figure 3a,b). Since the lowest intensities of π*-resonance are observed in the AEY spectra of the activated samples we conclude the modification occurred mainly in the outer surface layers. The states of carbon bonded with functional groups appear between π*- and σ*-resonances. The spectra of N-C sample have a peak at 288.3 eV, which can be attributed to oxygen-containing functional groups π*(C=O) or π*(C=C-NH_2_) [28,36,37]. The intensity of this peak increases and a new peak at 287.3 eV arises in the spectra of N-Cw and N-Ca. The latter peak corresponds to amine groups (C-NH_x_) or aliphatic carbon [38].

FTIR spectroscopy was additionally invoked to determine changes in the functional composition of N-doped carbon nanomaterial after the applied hydrothermal activations. In Figure 4, the FTIR spectra of the samples show a broad band centered at 3430 cm^−1^ corresponding to the stretching vibrations of O–H bonds from hydroxyl and carboxylic groups and adsorbed water [19]. The peak at 1400 cm^−1^ appears from O–H bending in carboxyl. The band at 1720 cm^−1^ is assigned to C=O stretching vibrations in carbonyl or carboxyl groups [39]. The C–O stretches of the hydroxyl and alkoxy groups also raise bands at 1285 cm^−1^ and 1020–1045 cm^−1^, respectively [40]. The peak at 1630 cm^−1^ is due to vibrations of aromatic C=C bonds and C=N bonds in the basal plane [24,41,42]. N–H stretching vibrations at 3220 cm^−1^ and N–H in-plane deformations at 1530 cm^−1^ [43] may overlap with other bands. C–N stretching vibrations can be detected at 1455 cm^−1^ [44]. A small peak at 880 cm^−1^ observed in the spectrum of N-C originates from C–O–C bonds [45]. Comparison of the FTIR spectra of the samples indicates (1) an increase of the content of C=O species in the row N-C < N-Cw < N-Ca, (2) conversion of C–O–C bonds present in N-C sample to –OH groups in the activated samples (N-Ca < N-Cw), (3) removal of alkoxy groups after the hydrothermal treatments, and (4) development of aliphatic C–H bonds with the vibrations at 2980, 2855, and 2920 cm^−1^ [46] in aqueous ammonia medium (sample N-Ca).

Nitrogen adsorption–desorption isotherms of the samples are presented in Figure 5a. According to the IUPAC classification [47], they are a combination of Type I and Type IV, which are associated with the adsorption in microporous (<2 nm in width) and mesoporous (2–50 nm in width) materials, respectively. The steep uptakes at very low P/P_0_ are associated with gas adsorption in micropores, while a hysteresis loop at P/P_0_ from ~0.5 to 1.0 arises due to the gas condensation in mesopores of the complex pore network. The values of specific surface area, pore-volume, and total volume of adsorbed nitrogen estimated from the isotherms are listed in Table 2. BET and DFT approaches give quite similar values of the specific surface area, indicating a small number of micropores in the samples. The DFT surface area values increase from 460 m^2^ g^−1^ for N-C to 494 m^2^ g^−1^ for N-Cw and 480 m^2^ g^−1^ for N-Ca. The DFT pore volume is 1.4 cm^3^ g^−1^ for N-C, 1.6 cm^3^ g^−1^ for N-Cw, and 1.3 cm^3^ g^−1^ for N-Ca. The N_2_ adsorption (V_ads_) increases after applied hydrothermal treatment in water and slightly decreases when the aqueous ammonia was used as the medium. It can be suggested that newly developed nitrogen-containing functional groups generated on defects and graphitic edges complicate the penetration of nitrogen molecules fully into microporous space. This assumption can explain the decrease in nitrogen absorption and pore volume for N-Ca sample.

Pore size distributions showed micropores of about 1 nm, small mesopores with a size of 4–5 nm, and large mesopores with a wide size distribution between 6 and 30 nm in the samples (Figure 5b). The mesopores arise as a result of the removal of template calcium oxide nanoparticles with hydrochloric acid treatment, and their size is determined by the temperature of calcium tartrate decomposition [24]. While the micropores more probably originate from atomic defects and incorporated nitrogen atoms in the graphitic lattice. The hydrothermal treatments increased amount of small mesopores (4–6 nm) in the samples and fraction of 0.9-nm micropores in N-Cw and 1.4-nm micropores in N-Ca. Moreover, the water-assisted treatment created the micropores with a size of 0.6 nm in N-Cw (Figure 5b). Based on previous works [48,49], the pores between 0.5 and 1 nm, close to the size of solvated hydrogen ions, give the largest contribution to an electric double-layer capacitance of carbon electrode. In the case of SIBs, the holes for the diffusion of solvated sodium ions should be more than 1 nm [50].

### 3.2. Electrochemical Properties

The results of electrochemical capacitive performances of the samples in 1M H_2_SO_4_ are shown in Figure 6. The specific capacitance of N-C sample was 125 F g^−1^ at a potential scan rate of 2 mV s^−1^ and it was reduced to 76 F g^−1^ at 20 mV s^−1^, 53 F g^−1^ at 50 mV s^−1^ and 34 F g^−1^ at 100 mV s^−1^ (Figure 6a). The sample N-Cw showed the capacitance of 140, 84, 55, and 35 F g^−1^ at the scan rates of 2, 20, 50, and 100 mV s^−1^, respectively. The sample N-Ca provided 135, 90, 60, and 37 F g^−1^ at 2, 20, 50, and 100 mV s^−1^, respectively, and exhibited the highest capacitance values at scan rates more than 2 mV s^−1^ among all samples. The developed pore structure, large specific surface area and functional groups of activated samples are responsible for fast ion diffusion and better wettability, therefore leading to higher capacitance and rate capability. Cycle-life tests of the activated samples at 100 mV s^−1^ showed a loss of initial specific capacitances less than ~5% during 2000 cycles (inset in Figure 6a).

Figure 6b,c shows the CV curves of the samples at scan rates of 5 and 10 mV s^−1^. The CV curves have a rectangular-like shape but the materials do not function as an ideal EDLC. The peaks appeared on the CV curves are related to the pseudocapacitance based on Faradaic reactions. The incorporated nitrogen and oxygen atoms typically take part in several redox reactions, thus leading to a rise of overlapped peaks on the CV curves. These reversible Faradaic processes occur with the proton-coupled electron transfer reaction of nitrogen- or oxygen-containing moieties due to the intercalation and absorption of H^+^ into material depth. Only carbonyl oxygen (C=O), pyridinic nitrogen (>N), and pyrrolic nitrogen (>N-H) are electrochemically active in acidic electrolytes [5,9,11,51]. At a scan rate of 5 mV s^−1^, the redox peaks from the reversible reduction/oxidation of carbonyl oxygen appear between 0.2 and 0.4 V [12], while the Faradaic process involving the pyridinic nitrogen gave a redox peak in the cathodic scans at a higher potential of ~0.7 V [52]. Thus, for the CV curves of N-C, the weak peaks at the potentials of 0.6–0.8 V in the cathodic scans can be mainly attributed to the pseudocapacitive contribution of N_pyr_ or N_hyd_. (Figure 6b,c). These peaks are more pronounced for the activated samples. For N-Cw, the redox peaks are observed at lower potentials of 0.2–0.4 V in the cathodic scans (Figure 6b,c). These peaks are attributed to Faradaic capacitance from the carbonyl groups involving protons of the aqueous acidic medium. This result agrees well with FTIR and XPS data, which detected the increasing in the concertation of carbonyl groups after the applied hydrothermal activations in water and aqueous ammonia media (Table 1, Figure 4). The more enhanced pseudocapacitive contribution was observed for N-Ca. In the CV curves, the two strong redox peaks observed at cathodic potentials of 0.5–0.6 V and 0.6–0.7 V (Figure 6b,c). Since the concentration of carbonyl and nitrogen functionalities increased in N-Ca, we suggest that the first peak originates from carbonyl groups, which are near protonated nitrogens and directly connected to an aromatic core. These groups can contribute to the electrochemical proton-coupled redox reaction giving shifted redox peaks in the CV curve. In accordance with the assumption of Wickramaratne et al., the redox reaction could involve pyridone-type nitrogen, where hydrogen migrates to neighbor carbonyl oxygen, and nitrogen bonds to H^+^ [53]. The second redox peak at higher potentials is corresponding to electrochemical reactions involving only nitrogen groups at graphene edges. Thus, capacitance values for N-Cw and N-Ca samples are higher than that for N-C owing to the presence of defects and oxygen- and nitrogen-containing functional groups, which increase the proton absorption in the acidic electrolyte.

Na-storage properties of the initial and activated N-doped carbon electrodes are demonstrated by galvanostatic charge–discharge, CVs and EIS (Figure 7 and Figure 8). Figure 7a shows the rate performance of N-doped carbon electrode. N-C sample had a reversible specific capacity of 141, 130, 111, 104, and 97 mAh g^−1^ at a current density of 0.05, 0.1, 0.25, 0.5, and 1 A g^−1^, respectively. For N-Cw, the specific capacities of 199, 194, 167, 152, and 135 mAh g^−1^ were achieved at the same current densities. N-Ca electrode material had the best cycling performance, with a specific capacity of 247, 240, 183, and 161 mAh g^−1^ at the same current densities. The capacity loss was less than ~5% for N-Cw and ~8% for N-Cw after 500 repeating cycles at a current density of 0.5 A g^−1^. Retention of capacity of 65–69% with an increase in current density from 0.05 to 1 A g^−1^ was found for the initial and activated samples. The observed increase in specific capacity by 40–50% for N-Cw and 60–85% for N-Ca could be caused by the formation of new active sites for Na-ion storage. These values are similar to those reported for various pure and N-doped carbonaceous materials, such as graphitic carbon, hard carbon, carbon fibers and biomass-delivered porous carbon [2,15,18,54,55,56,57]. The first cycle capacity loss of 1114 mAh g^−1^ for N-C, 1768 mAh g^−1^ for N-Cw, and 1803 mAh g^−1^ for N-Ca is attributed to the formation of a solid electrolyte interphase (SEI) layer and an irreversible reaction between the surface functional groups and Na^+^ ions (Figure 7a). Many heteroatom functional groups and defects on the surface of carbon shells favor surface-mediated Na-ion storage. The reduction in the first cycle capacity loss and an increase in the capacity retention can be achieved by optimizing the electrolyte composition through the use of specific additives, changing the parameters of the original material synthesis, or pretreatment of the carbon electrode using similar approaches developed for anode materials in LIBs [58,59,60].

Figure 7c shows galvanostatic profiles obtained for the three samples at the 55th charge/discharge cycles at the current density of 0.05 A g^−1^ when the formation of SEI layers was completed and the capacities had almost constant values. The profiles are typical for carbon electrodes and consist of two distinct regions: a plateau below 0.2 V corresponds to the intercalation/de-intercalation of Na^+^ ions into/from the interlayer space of pseudo-graphitic nanodomains and a slope region above 0.2 V, which is ascribed to reversible binding of Na^+^ ions with carbon defects, edge states or functional groups and their adsorption in the pores [61,62,63,64,65]. The slope region gives the main contribution of about 70% to the total Na-ion storage capacity of all samples. The main increase in the total capacity of the activated samples occurred due to the rise in the fraction of slope capacity since new defects and micropores act as new adsorption sites for Na^+^ ions. Dependencies of capacities at a potential below 0.2 V on the current density are shown in Figure 7b. The plateau capacity contributes no more than 30% of total values. It is in the range between 43–24 mAh g^−1^ for N-C, slightly increases to 56–30 mAh g^−1^ for N-Cw, and increases even more to 75–34 mAh g^−1^ for N-Ca. The more extended plateau capacities of activated samples are supposed to appear due to the presence of micropores developed during the hydrothermal treatments.

CV curves measured for the samples at a scan rate of 0.5 mV s^−1^ are presented in Figure 8a–c. A sharp peak at 0–0.2 V in the cathodic scans is related to the insertion of Na^+^ ions between the carbon layers, while the broad peak at 0.3 V in the reverse anodic scans corresponds to the extraction process [51]. The broad peaks at 0.6–0.8 V in the cathodic scans observed in the CV curves of all samples. These features are commonly not discussed, they can be attributed to the reversible binding of Na^+^ ions with the carboxylate units [66], nitrogen groups [14], or adsorption of Na^+^ ions above the center of monoatomic vacancies [67]. These peaks are more pronounced in the samples after the applied hydrothermal activation; therefore, we suggest that they arouse more likely due to the adsorption of Na^+^ ions in atomic defects than because of interaction with functional groups. It should be noted that we did not detect the redox peaks at 2.3/2.2 V, which Ye et al. attributed to the reversible reaction of sodium ions with C=O bonds [68].

EIS measurements were taken in order to understand why N-Cw and N-Ca electrodes exhibit such a superior electrochemical performance as compared to N-C one (Figure 8d). The Nyquist plots were modeled using an equivalent electrical circuit presented in the inset of Figure 8d. The ohmic resistance (R_1_) corresponds to the intersection of spectra with Re(Z) axis at 100 kHz. It includes the intrinsic resistance of the current collector and the electrolyte, as well as the contact resistance. A large semicircle in high frequency range corresponds to the charge transfer resistance (R_2_) and constant phase element (CPE_2_) for the SEI films. Another indistinct small semicircle and the sloping line in the lower frequency range are associated with the charge-transfer resistance (R_3_), the electric double-layer capacitance (C_3_) and the constant phase element for solid-state diffusion (CPE_3_) [69,70,71]. The resistance of SEI films (R_2_) and charge-transfer resistance (R_3_) calculated based on the fitted equivalent electrical circuits are 147 and 60 Ohm for N-C, 91 and 45 Ohm for N-Cw, and 119 and 46 Ohm for N-Ca. The lower R_2_ and R_3_ values for the activated samples indicates that the incorporation of carbonyl and nitrogen groups at the edges of graphene surface shells can improve the conductivity of SEI and carbon electrode material because of electron doping effect enhancing the electron transport during the electrochemical Na^+^ ion insertion/extraction processes. Moreover, the large surface area and pore volume can provide a sufficient electrode-electrolyte interface to storage Na^+^ ions, and thus improve the electrochemical performances. CPE_3_ (Z = 1/q(iω)^α3^) is defined by the exponent parameter α3, which describes ion diffusion behavior. According to the fitting results, α3 value is 0.75 for all samples. This value is larger than α = 0.5 characteristic of the Warburg impedance and it is closer to α = 1, which is typical for an ideal capacitor. Hence, Na^+^ ions diffuse only into fine surface layers of electrode carbon materials, where they form electric double-layers [71]. The very similar C_3_ and CPE_3_ elements describe the low frequency region of the Nyquist plots of all samples. Hence, the samples have comparable capacitive and ion diffusion behaviors, since the used activation procedures modified only surface shells in N-doped carbon material and did not cause significant changes in its morphology and structure.

## 4. Discussion

Structural changes that occurred in porous N-doped carbon material as the result of hydrothermal treatment in water with or without ammonia addition are illustrated in Figure 9 and collected in Table 3. The hydrothermal treatment caused both chemical modification and etching of the surface of carbon shells. FTIR, NEXAFS, and XPS data revealed that hydroxyl and carbonyl groups were attached to the surface of N-doped carbon after the treatment with water. Water acts as a source of hydroxyl functional groups yielding the attachment of these groups at the edges of graphene-like sheets, which, however, can be partially removed under hydrothermal conditions by the protonation process [72,73]. By cooling to room temperature, these neighboring hydroxyl groups can be further converted back to water molecules giving carbonyl groups on carbon. Niu et al. revealed that graphite oxide sheets were hydrothermally reduced in water at a temperature above 180 °C, and the lateral size of reduced sheets increased with the temperature [74].

The use of aqueous solutions of nitrogen-containing compounds in the hydrothermal process makes it possible to functionalize the carbon surface with nitrogen. For example, the wet chemical route is a very explicit and effective approach to fabricate N-doped graphene through the reactions of oxygenated graphite with thiourea [75], urea [76,77], sodium azide [78] and ammonia [79]. The hydrothermal reaction of ammonia solution with the carbon material is analogous to that in pure water. Aqueous ammonia dissociates to OH^−^ and NH_4_^+^ and H^+^ ions, which decorate the edges of graphene sheets. The total concentration of oxygenated species did not change in N-Cw, but the fraction of hydroxyl and carbonyl groups increased. According to FTIR and XPS data, N-Ca had the higher number of N_hyd_ and N_pyr_ edge groups. Additionally, NH_3_ can react with the oxygen functional groups such as carboxylic and hydroxyl species to form amide and amine groups [76]. These groups may be further dehydrated or decarbonylated with the formation of pyridinic, pyridonic, and pyrrolic nitrogen. The increased number of oxygen atoms double-bonded to carbon in N-Ca sample can be explained by their location near NH groups similar to pyridone (–NH–C=O).

The increased fraction of micropores and small mesopores in the activated samples indicates a partial distortion of carbon cage. Previously, it was reported that the hydrothermal treatment in water at 200 °C for 12 h resulted in the cutting of single-walled carbon nanotubes, which were firstly oxidized with nitric acid [80]. In comparison, the etching of non-modified carbon nanotubes by water vapor occurred at higher temperatures of 750–950 °C [81]. We suggest that cutting (breaking of carbon bonds) of the surface of N-doped carbon materials occurs as a result of chemical interaction of water with already existing oxygen-containing groups. The micropores of 0.6 nm were developed in water medium, while bigger micropores of 1.4 nm originated in aqueous ammonia solution. Moreover, both activation procedures led to an increase in the fraction of mesopores with a size of 3–6 nm due to perforation of carbon shells and opening access to previously closed internal cavities. The hydrothermal etching of pre-oxidized graphene and carbon nanotubes have been reported [7,80,82]. The cutting mechanism of oxidized graphene sheets suggested by Pan et al. involves the complete breakup of mixed chains composed of basal epoxy and more carbonyl groups under hydrothermal conditions [82]. In our case, the starting material has many oxygen-containing functional groups and defects, which promote etching of carbon shells under the hydrothermal treatment. The edges of the carbon sheets are reactive and readily interact with oxygen species [83,84,85].

Testing of N-doped carbon materials in the EDLC and SIB cells revealed an improvement of electrochemical performances of hydrothermally activated samples. Noteworthy, that as-grown N-doped carbon had a moderate BET surface area of 462 m^2^g^−1^ and a small fraction of micropores. Despite these structural characteristics, initial N-C material possessed quite high gravimetric capacitances in EDLC of 125 F g^−1^ at 2 mV s^−1^ and 76 F g^−1^ at 20 mV s^−1^, which correspond to volumetric capacitances of 6.2 F cm^−3^ and 3.7 F cm^−3^, respectively. The obtained capacitance values are comparable with those for nitrogen-containing carbon electrodes in 1M H_2_SO_4_ electrolyte, measured in similar three-electrode systems [10], and it is acceptable for practical application. We assume that the fairly good capacitance and rate capability of the N-C electrode are due to the prevailing mesoporosity (pore volume of 1.3 cm^3^ g^−1^), which provides quick access of electrolyte ions to the electrode surface especially at high rates [86,87].

The hydrothermal activated samples have increased electrochemical double-layer capacitance and better rate performance. These changes can be attributed to interconnected micro/mesopore-rich structure and larger specific surface, which give the plentiful interfaces for charge accumulation and short paths for ions/electrons. The specific capacitance increases with increasing the surface area and amount of both mesopores and micropores [49,88]. According to the current study, Faradaic reactions play an important role in the electric double-layer capacitances of the activated samples due to their increasing number of electrochemically active oxygen and nitrogen functional groups. The direct correlation between the pseudocapacitance and the nitrogen/oxygen content was revealed. The highest concentration of hydrogenated nitrogen and carbonyl groups was found in N-Ca sample, resulting in better pseudocapacitive behavior. Moreover, due to the higher number of micropores and mesopores in the activated samples, their enhanced specific surface areas of 511–518 m^2^ g^−1^ provided the increased specific electric double-layer capacitance. The observed synergetic effect of porosity and nitrogen functional groups in the ammonium-activated sample is in good agreement with the study reported by Hulicova-Jurcakova et al., which confirms that pyridinic, pyrrolic nitrogen, and quinone oxygen, presumably located in micropores bigger than 1 nm, have the most pronounced influence on the capacitance due to their pseudocapacitive contributions [48]. Hydrothermal treatment is an approach leading to the incorporation of nitrogen in carbons and improving their EDLC performance. The hydrothermal treatment of graphite oxide in an aqueous solution of urea yielded N-doped graphene with a high nitrogen content (up to 10 at%), which had a superior specific capacitance of 326 F g^−1^ in 6M KOH electrolyte at a current density of 0.2 A g^−1^ [76]. The hydrothermal treatment of an aqueous dispersion of graphite oxide by NH_3_ solution (0.01 vol.%) resulted in the synthesis of N-doped graphene aerogel with a nitrogen content of 6 at.% [79]. The aerogel, which underwent the hydrothermal treatment time for 45 min, had the highest gravimetric capacitance of 400 F g^−1^ at 50 A g^−1^ in 1M Na_2_SO_4_ electrolyte. For N-graphene production, Kim et al. used the microwave radiation for rapid heating of graphite oxide dispersion in an ammonia solution [89]. It appeared that the presence of oxygen-containing groups on the surface of carbon material is not required condition for hydrothermal etching of graphitic layer and nitrogen incorporation. Nitrogen-doped carbon materials prepared by hydrothermal carbonization of aminated tannin showed the highest specific capacitances up to 387.6 F g^−1^ at 2 mV s^−1^ [90,91].

The rather high Na-ion capacity of the initial N-C sample of 141 mAh g^−1^ at 0.05 A g^−1^ is due to the large number of mesopores, which facilitate the intercalation of Na^+^ ions [65,92]. In the case of the hydrothermally activated samples, the created defects open new pathways for penetration and accommodation of Na^+^ ions in an interlayer space of carbon shells. Micropores with a larger size of 1.4 nm formed in N-Ca compared to 0.9 nm in N-Cw sample. Apparently, the large micropores favor the intercalation of Na^+^ ions. Besides micropores, which provide spaces for storage of Na^+^ ions, the mesopores act as a host for electrolyte and facilitate the ion transport [92]. Yang et al. showed that the capacity of carbon with negligible micropores and abundant mesopores is 83% higher than the carbon with a few mesopores and abundant micropores [64]. Thus, both micropores and mesopores produced by the hydrothermal treatment facilitate Na^+^ ion intercalation. Nitrogen and oxygen groups also play role in the storage of Na^+^ ions. Our results agree well with previously reported data that the increased concentration of nitrogen, mainly pyridinic form, is one of the reasons for improving Na-ion storage capacity [14,15,18]. Piedboeuf et al. found that among the oxygen-containing functionalities only carbonyl groups resulted in the increased storage capacity of Li^+^ ions [93]. That also is in accordance with our results. Actually, N-Ca sample with the highest content of carbonyl and nitrogen groups showed the highest SIB performance.

## 5. Conclusions

A facile and economical hydrothermal treatment approach has been successfully used for chemical activation of as-grown porous N-doped carbon material for various electrochemical applications. By varying the hydrothermal treatment environment, the concentration and nature of functional groups, the number of defects, and pore size distribution can be changed. A treatment in water at 200 °C for 30 h resulted in the slight loss of nitrogen groups, mainly by removal of hydrogenated nitrogen, and the addition of carbonyl groups. A hydrothermal treatment in the ammonia solution at 147 °C for 30 h caused the incorporation of edge pyridinic and hydrogenated nitrogen groups and the addition of a larger number of carbonyl groups. In both media, the etching of outer carbon shells and the formation of micropores and small mesopores increased specific surface area and pore volume. For the sample activated in aqueous ammonia solution, larger micropores, and more surface defects were formed compared with the sample activated in water. Initial and activated porous N-doped carbon materials were used as electrode materials in EDLCs and SIBs and showed excellent electrochemical performances. The activation-induced micropores and nitrogen- and oxygen-containing functional groups enhanced the energy storage in EDLCs and SIBs due to appearance of more storage sites, faster ion transport and better conductivity of the electrode materials. The synergetic effect of large micropores with a size of 1.4 nm and an increased content of nitrogen and carbonyl surface edge groups in the ammonia-assisted treated N-doped carbon provided the maximum increase in the gravimetric capacitance by 8–15 F g^−1^ at 2–100 V s^−1^ in EDLC and the Na-ion storage capacity by 100 mAh g^−1^ in SIB. These results are significant for the further design of porous N-doped carbon materials for electrochemical applications.

## Figures and Tables

**Figure 1 nanomaterials-10-02163-f001:**
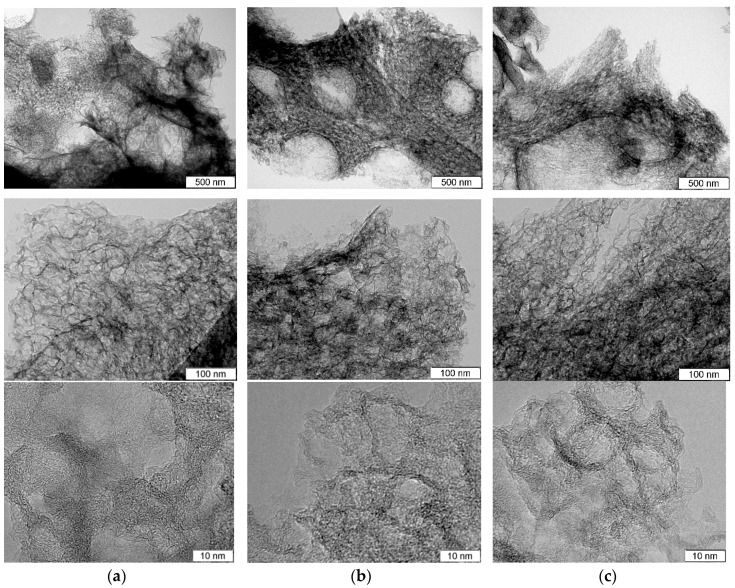
TEM images taken at various magnifications for (**a**) porous N-doped carbon material (N-C) and that after hydrothermal treatment in (**b**) water (N-Cw) and (**c**) ammonia solution (N-Ca).

**Figure 2 nanomaterials-10-02163-f002:**
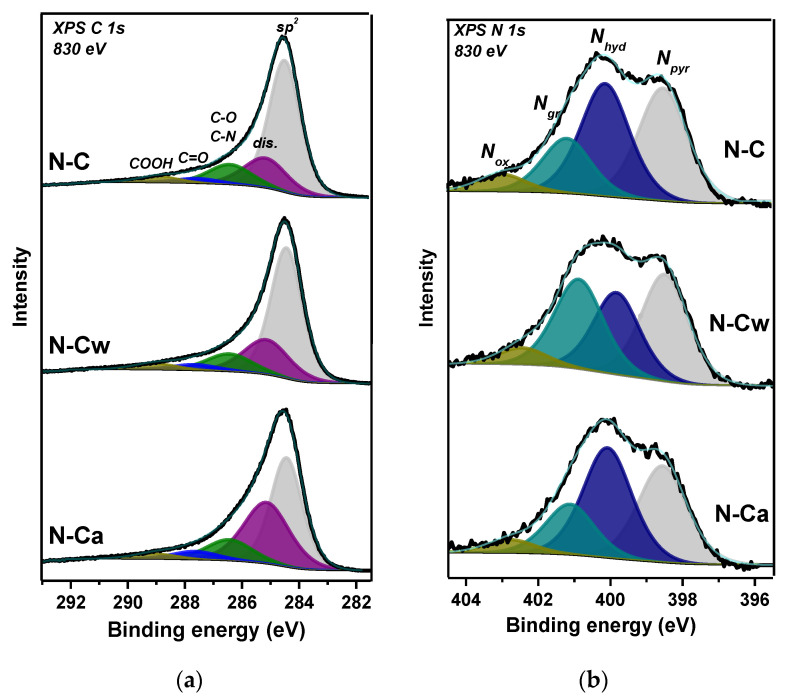
XPS (**a**) C 1s and (**b**) N 1s spectra of porous N-doped carbon material (N-C) and that after hydrothermal treatment in water (N-Cw) and ammonia solution (N-Ca).

**Figure 3 nanomaterials-10-02163-f003:**
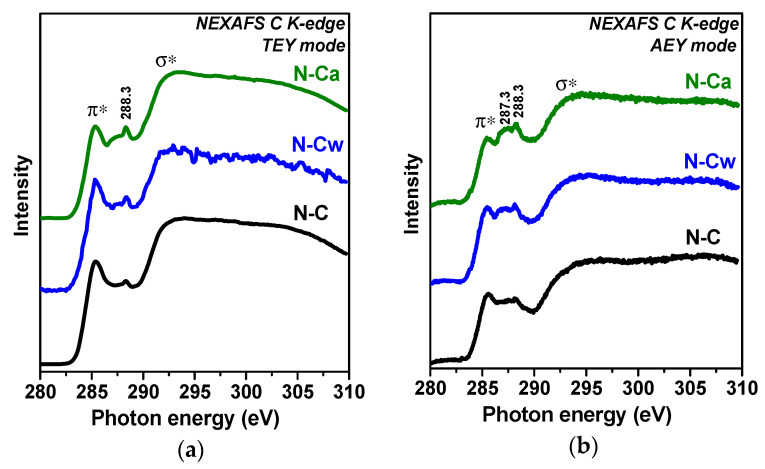
XPS (**a**) C 1s and (**b**) N 1s spectra of porous N-doped carbon material (N-C) and that after hydrothermal treatment in water (N-Cw) and aqueous ammonia solution (N-Ca).

**Figure 4 nanomaterials-10-02163-f004:**
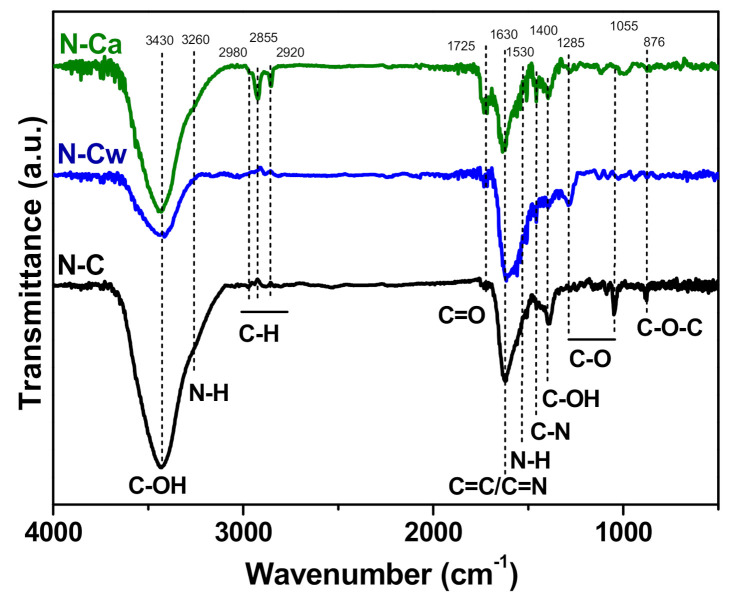
FTIR spectra of porous N-doped carbon material (N-C) and that after hydrothermal treatment in water (N-Cw) and ammonia solution (N-Ca).

**Figure 5 nanomaterials-10-02163-f005:**
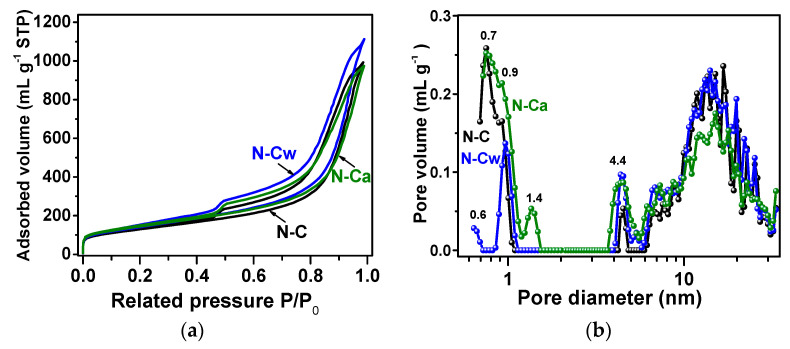
(**a**) Nitrogen adsorption–desorption isotherms measured at 77 K; (**b**) DFT (Density Functional Theory) pore size distributions for N-doped carbon material (N-C) and that after hydrothermal treatment in water (N-Cw) and ammonia solution (N-Ca).

**Figure 6 nanomaterials-10-02163-f006:**
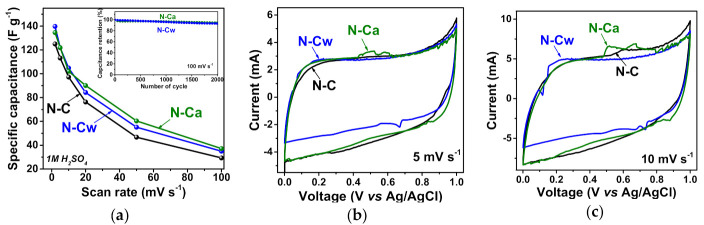
(**a**) Specific capacitances and CV curves at scan rates of (**b**) 5 and (**c**) 10 mV s^−1^ of porous N-doped carbon material (N-C) and that after hydrothermal treatment in water (N-Cw) and ammonia solution (N-Ca). Inset in Figure 6a shows capacitance retention plots for N-C and N-Ca during 2000 cycles at 100 mV s^−1^.

**Figure 7 nanomaterials-10-02163-f007:**
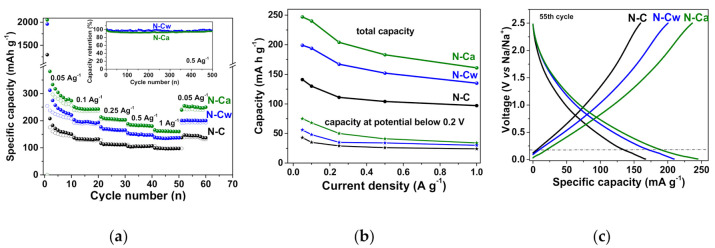
(**a**) Rate cycling performance of the samples and capacity retention plots for N-C and N-Ca during 500 cycles at a current density of 0.5 A g^−1^ (inset), (**b**) total capacity and plateau capacity below 0.2 V estimated from charge/discharge profiles at different current densities, (**c**) charge–discharge profiles of 55th cycles at a current density of 0.05 A g^−1^ for the initial sample (N-C) and that hydrothermally activated in water (N-Cw) and ammonia solution (N-Ca).

**Figure 8 nanomaterials-10-02163-f008:**
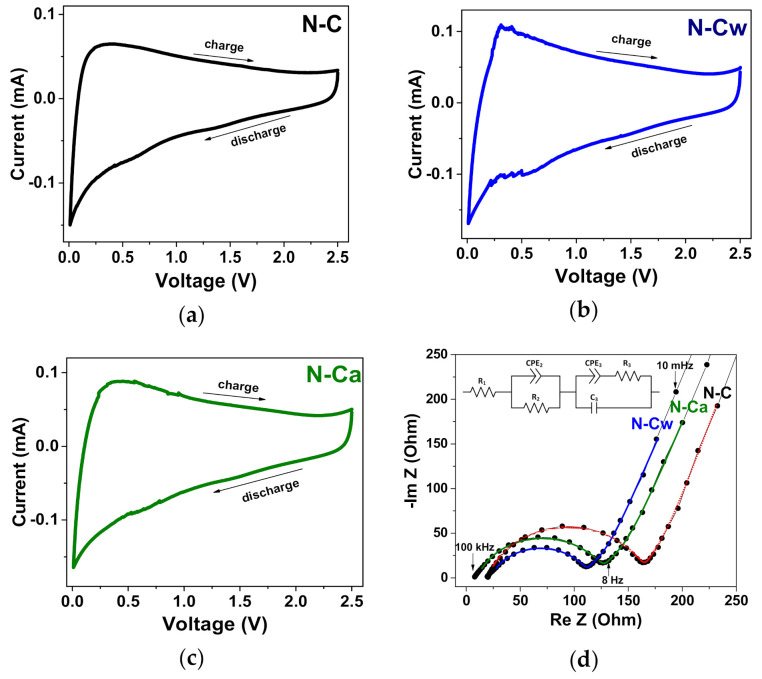
(**a**–**c**) CV curves measured at a scan rate of 0.5 mV s^−1^ and (**d**) electrochemical impedance spectra (EIS) curves for the initial sample (N-C) and that after hydrothermal treatment in water (N-Cw) and ammonia solution (N-Ca). The inset in Figure 8d shows an equivalent circuit used for the fitting of EIS curves.

**Figure 9 nanomaterials-10-02163-f009:**
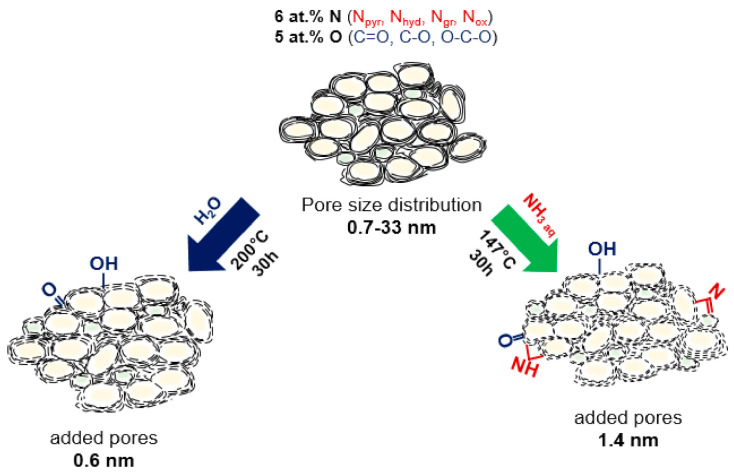
Schematic illustration of the structural evolution of N-doped carbon sheets in the hydrothermal process after the treatment in water and aqueous ammonia solution.

**Table 1 nanomaterials-10-02163-t001:** Energy, assignment and related area of the components in XPS C 1s and N 1s spectra.

Energy, eV	C 1s XPS					N 1s XPS			
	284.5	285.2	286.5	287.7	288.8	398.5	400.1	401.2	402.3
Assignment	sp^2^	dis.	C-O/C-N	C=O	COOH	N_pyr_	N_hyd_	N_gr_	N_ox_
Related area, % N-doped carbon	61	19	13	3	4	38	38	18	6
After treatment in H_2_O	58	23	12	4	4	36	27	31	6
After treatment in NH_3_(aq.)	44	36	12	5	4	36	41	19	5

**Table 2 nanomaterials-10-02163-t002:** The parameters of porous structure of samples under investigation.

Sample	Specific Surface Area, m^2^·g^−1^		V_pore_, cm^3^·g^−1^		Vads ^a^, cm^3^(STP)·g^−1^
	BET	DFT	Total ^a^	DFT	
N-doped carbon material	462	440	1.30	1.41	840
After treatment in H_2_O	511	494	1.45	1.56	937
After treatment in NH_3_(aq.)	518	480	1.19	1.32	768

^a^ measured at P/P_0_ = 0.95.

**Table 3 nanomaterials-10-02163-t003:** Concentration of electrochemically active groups, volume of micropores and small mesopores, and capacitive characteristics of the samples under investigation.

Sample	Concentration, At.%			Pore Volume, 10^−2^ cm^3^·g^−1^		Electrochemical Double-Layer Capacitors (EDLCs) Capacitance At 20 mV s^−1^, F g^−1^	Na^+^ Storage Capacity At 0.05 A g^−1^, mA h g^−1^
	C=O	N_pyr_	N_hyd_	0–2 nm	2–6 nm		
N-doped carbon material	3	2	2	3	1	76	141
After treatment in H_2_O	4	2	1	2	3	84	199
After treatment in NH_3_(aq.)	5	3	3	5	6	90	247

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
