# Peer review of "Hydrothermal Activation of Porous Nitrogen-Doped Carbon Materials for Electrochemical Capacitors and Sodium-Ion Batteries"

_nanomaterials, 2020, doi:10.3390/nano10112163_

Round 1

Reviewer 1 Report

Fedoseva et al. Submitted a manuscript to Nanomaterials focused on the effect of hydrothermal treatment in porous n-doped carbons. Two applications are presented: electrochemical capacitors and Na-Ion batteries.

Prepared carbon materials have a moderate to low surface area and a high mesoporous volume, so their application as supercapacitor electrode is not really suitable. The specific capacitances are low if the capacitances were reported in volume rather than weight, the results would be even worse. This issue needs to be addressed in the conclusion that should critically examine the results.

Some important references on N-doping at hydrothermal conditions are missing (10.1016/j.carbon.2012.07.027 ;10.1016/j.carbon.2015.03.038) and should be included

The importance of surface oxygen groups in Li-Ion batteries has recently been evaluated (10.1021/acsami.0c08297), this manuscript should be included in the discussion of Na+ insertion/de-insertion capacities. Is the evolution of capacity with oxygen groups similar or not when the results on Na-ion are compared to Li-Ion batteries?

Author Response

We thank the Reviewer for evaluation of our work and below provide a response to each comment.

Point 1: Prepared carbon materials have a moderate to low surface area and a high mesoporous volume, so their application as supercapacitor electrode is not really suitable. The specific capacitances are low if the capacitances were reported in volume rather than weight, the results would be even worse. This issue needs to be addressed in the conclusion that should critically examine the results.

Response 1: We thank the Reviewer for this remark. To clarify this issue, we added some sentences in the revised manuscript in the part Discussion: “Note, that as-grown N-doped carbon has a moderate BET surface area of 462 m2g-1 and small fraction of micropores. Despite these structural characteristics, initial N-C material possesses quite high gravimetric capacitances in EDLC of 125 F g-1 at 2 mV s-1 and 76 F g-1 at 20 mV s-1, which correspond to volumetric capacitances of 6.2 F cm-3 and 3.7 F cm-3, respectively. The obtained capacitance values are comparable with those for nitrogen-containing carbon electrodes in 1M H2SO4 electrolyte, measured in similar three-electrode systems [10], and it is acceptable for practical application. We assume that the fairly good capacitance and rate capability of the N-C electrode are due to the prevailing mesoporosity (pore volume of 1.3 cm3 g-1), which provides quick access of the electrolyte ions to the surface of the electrode particularly at high rates [88, 89].”

Point 2. Some important references on N-doping at hydrothermal conditions are missing (10.1016/j.carbon.2012.07.027 ;10.1016/j.carbon.2015.03.038) and should be included.

Response 2: We are thankful for the additional references suggested by the Reviewer. They were added to the end of Discussion: “Nitrogen-doped carbon materials prepared by hydrothermal carbonization of aminated tannin showed the highest specific capacitances up to 387.6 F g-1 at 2 mV s-1 [92, 93].”

Point 3: The importance of surface oxygen groups in Li-Ion batteries has recently been evaluated (10.1021/acsami.0c08297), this manuscript should be included in the discussion of Na+ insertion/de-insertion capacities. Is the evolution of capacity with oxygen groups similar or not when the results on Na-ion are compared to Li-Ion batteries?

Response 3: Many thanks for the useful reference. The results obtained in that paper was discussed in the last paragraph of the part Discussion: “Marie-Laure et al found that among the oxygen-containing functionalities only carbonyl groups resulted in the increased storage capacity of Li+ ions [95]. That also is in accordance with our results. Actually, N-Ca sample with the highest content of carbonyl and nitrogen groups showed the highest SIB performance.”

Reviewer 2 Report

Fedoseeva et al have activated porous nitrogen-doped carbon materials for electrochemical capacitors and sodium-ion batteries. I highly appreciate the authors for their effort in discussing the results and the observations. This manuscript can be accecpted in the presigious Nanomaterials journal after addressing the following minor comments.

  1. Introduction section is bit lengthy. Can be shortned to give more impact on the results and discussion section.
  2. Authors have not discussion about the electrochemical stability of the carbon materials. This is an important information and should be included in the manuscript.

Author Response

We are grateful to the Reviewer for a positive feedback on the reported results and valuable comments and suggestions. According to the above suggestions, the following changes were made in the manuscript.

Point 1: Introduction section is bit lengthy. Can be shortened to give more impact on the results and discussion section.

Response 1: The introduction was rewritten more concisely.

Point 2: Authors have not discussion about the electrochemical stability of the carbon materials. This is an important information and should be included in the manuscript.

Response 2: The cycling stability tests of activated samples in EDLC and SIB were added as insets in Figure 6a and Figure 7a. The obtained data were described in the part Results as: “Cycle-life tests of the activated samples at 100 mV s-1 showed the loos of initial specific capacitances less than ~5% during 2000 cycles (inset to Fig. 6a)” and “The capacity loss was less than ~5% for N-Cw and ~8% for N-Cw after 500 repeating cycles at a current density of 0.5 A g-1.”

Reviewer 3 Report

The manuscript presents the study on the synthesis and characterization of N-doped carbon materials and their application as an active electrode material for supercapacitors and an electrode material for sodium-ion battery (NIB). The synthesis was relied on deposition of acetonitrile vapors  at 750°C on calcium tartrate-based CaO nanoparticles to obtain N-doped carbon followed by its hydrothermal treatment at 200°C in water and aqueous ammonia solution. The manuscript cannot be accepted for publication in high-ranked journal as Nanomaterials is because of the following reasons.

No novelty in the synthesis approach for obtaining N-doped carbon materials. The synthesis of N-doped carbons was reported by authors in Physica Status Solidi 251 (2607-2612) 2014.

Authors proposed hydrothermal treatment as an activation method for developing porous structure of carbons. It is hardly to accept. Physical activation using steam or CO2 at high temperatures (800-900°C) and chemical activation using H3PO4, KOH and ZnCl2 as activating agents at temperatures of 450-800oC are widely used for obtaining carbons with developed porous structure; some of them are applied on industrial scale. The mechanism of pore development in case of each method is well-known. Hydrothermal treatment does not belong to approaches leading to porosity development. It can be even supported by the results obtained in this work (Table 2). The specific surface area increased due to hydrothermal treatment to very small extent. Moreover, the total pore volume was decreased after hydrothermal treatment in ammonia solution. Also the data in Table 3 confirmed that the changes in the micropore and mesopore (with a size of 2-6 nm) volumes are insignificant. Summarizing, we cannot consider hydrothermal treatment as a method of porosity development.

The BET surface area of the N-doped carbons is between 480 and 518 m2/g (according to DFT 440-480 m2/g) which is extremely low in terms of their application as electrode material for supercapacitors. Therefore, such materials should not be considered in the study on the electrochemical capacitors. Moreover, the synthesized carbons are characterized by very low micropore volume (0.03-0.06 cm3/g) which indicates again that they are not suitable for EDL capacitors. Indeed, the specific capacitance of carbons was low, i.e. 125-140 F/g (2 mV/s). The electrochemical behavior was determined in 1M H2SO solution. It is well-known that the higher surface area of the carbon electrode, the more energy can be stored in its more developed electrode/electrolyte interface for the formation of the electric double layer. The commercial activated carbons for EDL capacitor have the BET surface area over 1500 m2/g.

Having the structural and textural characteristics of the synthesized carbons it is easy to predict that the synthesized carbons are also not suitable for application as electrode material for Na-ion batteries. Huge irreversible capacity (1114-1803 mAh/g) strongly confirms this statement. Moreover, the presence of heteroatom in the carbon structure is not desirable in this application. As far as the porous texture is concerned, the high volume of narrow micropores with a size lower than 1nm are preferable in carbons used for NIBs. This is not the case.

Author Response

We are thankful to the Reviewer for valuable comments and critical assessment of the manuscript. The main comment is related to the fact that obtained N-doped carbon materials are not suitable for industrial application as electrode material of electrochemical capacitors and Na-ion batteries. We agree with this statement and we do not in any way claim that our samples, as they are now, can be used industrially for the production of electrochemical devices. We want to note that our article describes exclusively scientific results in the field of material science and Nanomaterials journal suites well for publishing research papers. The development of new synthetic approaches for design of materials with improved properties, including electrochemical, is topical problem of nanotechnology. We highlight our point of view in the part Introduction: “One of the major challenges in the development of electrochemical devices is electrode materials with outstanding performances. In this regard, extensive research is needed to establish the structure-property relationships in electrode material and to search for new approaches to achieve the desired material properties.”

Point 1: No novelty in the synthesis approach for obtaining N-doped carbon materials. The synthesis of N-doped carbons was reported by authors in Physica Status Solidi 251 (2607-2612) 2014.

Response 1: The Reviewer is right, that the method of the synthesis of porous N-doped carbon material has been described in our previous paper, which was cited in present manuscript as reference 24. However, we did not test in that paper the obtained material for electrochemical applications. Moreover, in present manuscript the synthesis of the initial material was modified as compared to Ref. 24.  Here we used calcium tartrate without doping with iron and lower the temperature of the synthesis to 750 °C instead of 800 °C. The details of the synthesis were described in the Experimental section: “Porous N-doped carbon material (denoted as N-C) was synthesized by the CVD method using conditions described in details elsewhere [24]. Decomposition of calcium tartrate (CaC4H4O6) at 750 °C yielded carbon-coated CaO nanoparticles, which were used for the deposition of acetonitrile (CH3CN) vapors at the same temperature, leading to the formation of graphitic layers.”

In present manuscript, we do not claim that we have developed a new method for the synthesis of N-doped carbon, but focus on issues related to its hydrothermal modification and the effect of this modification on the electrochemical properties of obtained materials.

Point 2: Authors proposed hydrothermal treatment as an activation method for developing porous structure of carbons. It is hardly to accept. Physical activation using steam or CO2 at high temperatures (800-900°C) and chemical activation using H3PO4, KOH and ZnCl2 as activating agents at temperatures of 450-800oC are widely used for obtaining carbons with developed porous structure; some of them are applied on industrial scale. The mechanism of pore development in case of each method is well-known.

Response 2: We fully agree with the Reviewer that the abovementioned activation methods develop a porous structure of carbon materials, they are used in the industry. Despite the applied technologies, inquisitive scientific researchers should not be satisfied with what has already been achieved! These methods are not universal for all carbon materials and have a number of disadvantages. Firstly, the activating reagents are very aggressive and polluting. Secondly, a lot of carbon material and incorporated nitrogen can be lost during their processing. Thus, the development of new environmentally friendly and resource-saving protocols must not be ignored. This point is clarified in the part Introduction: “Physical activation by CO2 and chemical activation by KOH, ZnCl2, and acids are widely used to create pores in carbon materials [19-23]. However, such treatments require high temperatures and/or aggressive reagents, which cause dramatic loss of carbon material and nitrogen incorporated in the carbon lattice [20, 23] that may not satisfy the required characteristics of electrode materials.”

Note, that we used the activation procedure with KOH for our N-doped carbon material. Specific surface area of N-doped carbon increased from 527 to1800 m2 g-1 due to development of numerous micropores, but concentration of nitrogen decreased to 1 at%. The KOH-activated materials showed an increase in capacitance only at scan rates lower than 10 mV s-1 (see Figure), but did not change the capacitance at higher scan rates.

Figure. Specific capacitances at scan rates of 2-100 mV s-1 of N-doped carbon material (N-C) and that after activation by KOH (left) and hydrothermal treatments (right).

The hydrothermal treatment described in the manuscript leads to an increase in capacitance of the sample in the range of high scan rates. Moreover, the Na-ion capacity of the treated materials was also increased.

Point 3: Hydrothermal treatment does not belong to approaches leading to porosity development. It can be even supported by the results obtained in this work (Table 2). The specific surface area increased due to hydrothermal treatment to very small extent. Moreover, the total pore volume was decreased after hydrothermal treatment in ammonia solution. Also, the data in Table 3 confirmed that the changes in the micropore and mesopore (with a size of 2-6 nm) volumes are insignificant. Summarizing, we cannot consider hydrothermal treatment as a method of porosity development.

Response 3: Our main conclusion is not that the hydrothermal treatment is one of the approaches leading to the development of high porosity. However, the increase in porosity is difficult to deny as data in Table 2 show. According to nitrogen adsorption technique, BET surface area increased by 49 and 56 m2 g-1 after the treatment in water and ammonia solution, respectively. We suggest creation of small atomic defects, which may be more easily detected by CO2 adsorption probably. This suggestion confirmed by suppressed π*-resonance in the NEXAFS CK-spectra of threated sample (Figure 3). It means that surface carbon layers become more etched after hydrothermal treatment. These atomic defects are new small pores and electrochemical testing gives a stronger response to the presence of these defects than N2 adsorption.

The hydrothermal treatment induces new functional groups, which causes better electrical conductivity of the material, and increases the fraction of mesopores, which improve ion transport. The advantages of the used treatments are summarized in the part Conclusion: “The activation-induced micropores and nitrogen- and oxygen-containing functional groups enhanced the energy storage in EDLCs and SIBs due to appearance of more storage sites, faster ion transport, and better conductivity of the electrode materials.”

Point 4: The BET surface area of the N-doped carbons is between 480 and 518 m2/g (according to DFT 440-480 m2/g) which is extremely low in terms of their application as electrode material for supercapacitors. Therefore, such materials should not be considered in the study on the electrochemical capacitors. Moreover, the synthesized carbons are characterized by very low micropore volume (0.03-0.06 cm3/g) which indicates again that they are not suitable for EDL capacitors. Indeed, the specific capacitance of carbons was low, i.e. 125-140 F/g (2 mV/s). The electrochemical behavior was determined in 1M H2SO solution. It is well-known that the higher surface area of the carbon electrode, the more energy can be stored in its more developed electrode/electrolyte interface for the formation of the electric double layer. The commercial activated carbons for EDL capacitor have the BET surface area over 1500 m2/g.

Response 4: The high surface area of carbon material is due to large fraction of micropores. These micropores create many sites for ion adsorption, however, they make carbon material amorphous-like. As a result, the material has a low electrical conductivity and became not suitable for electrochemical applications. Mesopores in carbon material play important role for mass-transport of electrolyte. The carbon samples enriched with mesopores have the highest electrochemical double-layer capacitance. We disagree with the statement that the specific capacitance of carbons is low, i.e. 125-140 F/g (2 mV/s). The obtained values agree well with results reported for other N-doped carbon materials with similar concentration of nitrogen [10.1039/C5TA08620E.]. This point was discussed in the part Discussion: “Note, that as-grown N-doped carbon has a moderate BET surface area of 462 m2g-1 and small fraction of micropores. Despite these structural characteristics, initial N-C material possesses quite high gravimetric capacitances in EDLC of 125 F g-1 at 2 mV s-1 and 76 F g-1 at 20 mV s-1, which correspond to volumetric capacitances of 6.2 F cm-3 and 3.7 F cm-3, respectively. The obtained capacitance values are comparable with those for nitrogen-containing carbon electrodes in 1M H2SO4 electrolyte, measured in similar three-electrode systems [10], and it is acceptable for practical application. We assume that the fairly good capacitance and rate capability of the N-C electrode are due to the prevailing mesoporosity (pore volume of 1.3 cm3 g-1), which provides quick access of the electrolyte ions to the surface of the electrode particularly at high rates [88, 89].”

Point 5: Having the structural and textural characteristics of the synthesized carbons it is easy to predict that the synthesized carbons are also not suitable for application as electrode material for Na-ion batteries. Huge irreversible capacity (1114-1803 mAh/g) strongly confirms this statement. Moreover, the presence of heteroatom in the carbon structure is not desirable in this application. As far as the porous texture is concerned, the high volume of narrow micropores with a size lower than 1nm are preferable in carbons used for NIBs. This is not the case.

Response 5: We agree that the produced materials and electrolyte as they are can not be used for industrial application in SIBs. But this paper is only research study and the problem of how to reduce high irreversible capacitance in the first cycle will be in our future consideration. Approaches, which can be used for decrease of the irreversible capacity of the materials are described in the part Results: “The first cycle irreversible capacity loss of 1114 mAh g-1 for N-C, 1768 mAh g-1 for N-Cw, and 1803 mAh g-1 for N-Ca is attributed to the formation of a solid electrolyte interphase (SEI) layer and an irreversible reaction between the surface functional groups and Na+ ions (Fig. 7a). Many of heteroatom functional groups and defects on the surface of carbon shells favor surface-mediated Na-ion storage. The reduction in the first cycle irreversible capacity loss and an increase in the capacity retention can be achieved by optimizing the electrolyte composition through the use of specific additives, changing the parameters of the original material synthesis, or pretreatment of the carbon electrode using the similar approaches developed for anode materials in LIBs [59-61].”

Nevertheless, there are numerous works devoted to the positive influence of mesopores and functional groups on sodium-ions storage by carbon materials. Our research results based on N-doped carbon material do not contradict with the previously published ones. As examples, Yang et al claimed the micropores hinder ion diffusion and hardly ever accommodate Na+ ions, while mesopores facilitate Na+ ion intercalation [10.1016/j.jcis.2020.01.085]. The role of oxygen and nitrogen containing groups in the intercalation/deintercalation of alkali metal ions are fully described in [10.1039/C9TA04699B, /10.1039/C9CC08493B]. This issue is described in the last paragraph of the part Discussion: “The rather high Na-ion capacity of the initial N-C sample of 141 mAh g-1 at 0.05 A g-1 is due to the large number of mesopores which facilitate intercalation of Na+ ions [65, 94]. In the case of the hydrothermally activated samples, the created defects open new pathways for penetration and accommodation of Na+ ions in an interlayer space of carbon shells. Micropores with a larger size of 1.4 nm formed in N-Ca compared to 0.9 nm in N-Cw sample. Apparently, the large micropores favor the intercalation of Na+ ions. Besides micropores, which provide spaces for storage of Na+ ions, the mesopores act as a host for electrolyte and facilitate the ion transport [94]. Yang et al. showed that the capacity of carbon with negligible micropores and abundant mesopores is 83% higher than the carbon with a few mesopores and abundant micropores [65]. Thus, both micropores and mesopores produced by the hydrothermal treatment facilitate Na+ ion intercalation. Nitrogen and oxygen groups also play a role in the storage of Na+ ions. Our results agree well with previously reported data that the increased concentration of nitrogen, mainly pyridinic form, is one of the reasons for improving Na-ion storage capacity [14,15, 18]. Marie-Laure et al found that among the oxygen-containing functionalities only carbonyl groups resulted in the increased storage capacity of Li+ ions [95]. That also is in accordance with our results. Actually, N-Ca sample with the highest content of carbonyl and nitrogen groups showed the highest SIB performance.”

Round 2

Reviewer 1 Report

Concerning response 3: "Marie-Laure" is a first name and not a family name,It should be written "Piedboeuf et al."

In the title of ref 92, it should be written "tannins" and not "tanninas"

Reviewer 3 Report

The manuscript has been improved taking into account my comments. It can be accepted for publication.